# Metallogenic Mechanism and Geodynamic Background of the Chang'an Chong Cu-Mo Deposit in Southern Ailaoshan Tectonic Belt: New Evidence from Garnet U-Pb Dating and In-Situ S Isotope

**Bin Sun** [1] , **Yi Liu** [1,*], **Yongfeng Yan** [1,*], **Lei Ye** [2] and **Gang Chen** [3]

1. Faculty of Land Resources Engineering, Kunming University of Science and Technology, Kunming 650093, China
2. Yunnan Dianxiang Mining Industry Consultants Co., Ltd., Kunming 652017, China
3. Institute of Mineral Resources, Chinese Academy of Geological Sciences, Beijing 100037, China
* Correspondence: yiliu@kust.edu.cn (Y.L.); yanyongfeng@kust.edu.cn (Y.Y.)

**Abstract:** The Chang'an Chong Cu-Mo deposit is located in the Chang'an Cu-Mo-Au ore cluster in the southern Ailaoshan tectonic belt in southwestern China. There are six intrusive bodies in the mining area, among which the No.II intrusive body is the largest and most closely related to Cu-Mo mineralization of skarn. The No. 1 main orebody is composed of the No. 1 copper orebody and No. 1 molybdenum orebody, which are distributed in parallel with similar shapes. In this paper, the age of skarn is determined by the LA-SF-ICP-MS U-Pb dating of garnet, and it is nearly consistent with the age of alkaline porphyry in this region (41–32 Ma). Compared with the U-Pb age of zircon from the ore-bearing porphyry and the Re-Os age of molybdenite, the U-Pb age of garnet was consistent with them within error, indicating that they were the same mineralization event, which further proves that the porphyry-skarn Cu-Mo-Au mineralization event along the Ailaoshan-Red River fault zone mainly occurred at 38~32 Ma. In-situ S isotope results show that the $\delta^{34}S$ mean values of disseminated pyrite (PyI), pyrite of sulfide veins (PyII) and chalcopyrite (Ccp) in the main mineralization period are 2.35‰, 3.60‰ and 0.55‰, respectively. These $\delta^{34}S$ values are similar to those of magma and slightly enriched in $\delta^{34}S$, and the $\delta^{34}S$ value of chalcopyrite is mainly concentrated near 0‰, so it can be considered that the S of the ore-forming fluid came from magmatic-hydrothermal fluids. Based on the comprehensive analysis of the regional metallogenic background, deposit chronology and isotope geochemistry, it is concluded that the Chang'an Chong Cu-Mo deposit was formed in an intra-plate post-collision strike-slip environment.

**Keywords:** Chang'an Chong Cu-Mo deposit; garnet U-Pb dating; in-situ S isotope; the metallogenic mechanism

## 1. Introduction

The Ailaoshan-Red River fault zone is an NW-NNW-trending deep fault in the Sanjiang region of southwestern China, which was formed by the intense deformation of the Indian plate and Eurasian plate after the Cenozoic collision. It is accompanied by a large area of ~41–32 Ma alkaline porphyry outcrop [1–6]. The alkaline porphyry is more than 1000 km in length and ~50–80 km in width. A series of large and medium-sized Cu-Mo-Au metal deposits are developed, such as Beiya Au, Chang'an Au, Chang'an Chong Cu-Mo, Machangqing Cu-Mo, Tongchang Cu-Mo, etc. This alkaline porphyry is one of the important porphyry-skarn copper metallogenic provinces and metallogenic prospects in China. Therefore, typical porphyry-skarn Cu-Mo-Au deposits related to alkaline porphyry can be used as the typical examples to study regional tectonic-magmatic-mineralization processes, continental collision mechanisms and deep processes [7]. The series of deposits in the southern segment of the Ailaoshan-Red River fault zone are called the Chang'an Cu-Mo-Au ore

cluster by previous studies [8], and the Chang'an Chong Cu-Mo deposit is one of the typical deposits. The proven resource of Cu metal is 10,308 t, with an average grade of 1.59%; the amount of Mo metal is 380 t, with an average grade of 0.2%. The Chang'an Chong Cu-Mo deposit is currently a small one but adjacent to the famous large-scale Chang'an Au deposit, and it has the potential to find more resources in the future. Previous studies confirmed that magmatic activities mainly occurred at $34.5 \pm 0.3$ Ma and $21.7 \pm 0.3$ Ma in this area, but the metallogenic age needs to be further determined [8,9]. The $\delta^{34}$S values range from 0.2‰ to 1.5‰ [9], indicating the characteristics of mantle-derived sulfur. The S isotope analysis of metal sulfides in this area is determined by the monomineral powder method. However, a sulfide often contains other sulfide minerals, and the selected mineral particles cannot be fully guaranteed to be pure single minerals. As a result, the obtained $\delta^{34}$S value cannot accurately represent the true value of a single sulfide mineral, which in turn affects the tracing of S sources.

Garnet, a common mineral in skarn, has a high sealing temperature of U-Pb isotope system > 850 °C [10] and is used for radioisotope dating [11,12]. In recent years, thanks to the development of laser-denuded inductively coupled plasma mass spectrometry (LA-ICP-MS) analysis technology, it has been possible to successfully obtain the high-resolution dating of garnet, a low U mineral. Now, the dating of samples with U contents lower than 1 ppm can be evaluated [13]. Notably, the detailed U mapping of garnet illustrates a uniform distribution, indicating that U occurs within the garnet crystal lattice. U-Pb ages of garnet therefore represent the crystallization ages of the rock, which are of critical importance to establish a temporal link with ore-related intrusive bodies [13–25]. Sulfide minerals with complex structures can record the chemical properties and the evolution process of ore-forming fluids, so the S isotope analysis can be used to analyze the geochemical environment and source of ore-forming materials [26,27], and track the changes in sulfur sources accurately. In this paper, the LA-SF-ICP-MS technique was used to date the original location of garnet in the Chang'an Chong Cu-Mo deposit, combined with the study of mineralogy to constrain the metallogenic age. At the same time, the LA-MC-ICP-MS technique was used to determine the S isotopes of sulfides (pyrite and chalcopyrite) to ascertain the source of ore-forming materials, and to try to provide some indications for the genesis of the deposit.

## 2. Regional Geological Setting

The Sanjiang Tethys structural belt, which is approximately 1000 km long and ~20–80 km wide in southwest China [28], is distributed in the NW-SE direction with a broomlike shape. To the south, the belt crosses Vietnam to the South China Sea, and is bordered by the Qamdo-Simao block to the southwest and the North China plate to the northeast (Figure 1a,b). Its formation and evolution process are accompanied by the strong collision and subduction of the Eurasian and Indian plates [29–32].

The Ailaoshan tectonic belt is an important part of the Sanjiang Tethys tectonic belt, and the Chang'an Chong Cu-Mo deposit is located in the south of the Ailaoshan tectonic belt.

The strata in the area are distributed from Proterozoic to Quaternary. However, due to the extremely strong tectonic activity, the stratigraphic age is relatively chaotic, so it is impossible to establish a complete system of stratigraphic sequence.

Cenozoic magmatic rocks are mainly developed in the area. Influenced by Himalayan orogeny, during the Eocene to early Oligocene, large-scale shear occurred along the Ailaoshan tectonic belt, and a large number of potassic magma upflowed, forming the ~41~32 Ma potassic igneous belt [33]. The felsic intrusive bodies mainly exist in syenite porphyry, adakite granite and quartz monzonite, and the end-members are generally characterized by high silicon, rich in potassium, alkali and calcium, which are closely related to a series of Cu-Au-Mo deposits in the area. During the late Oligocene to early Miocene, the tectonic system changed from shear to a regional extensional environment, and the potassium-rich calc-alkalic magmatic rocks mainly developed at this time [34–36].

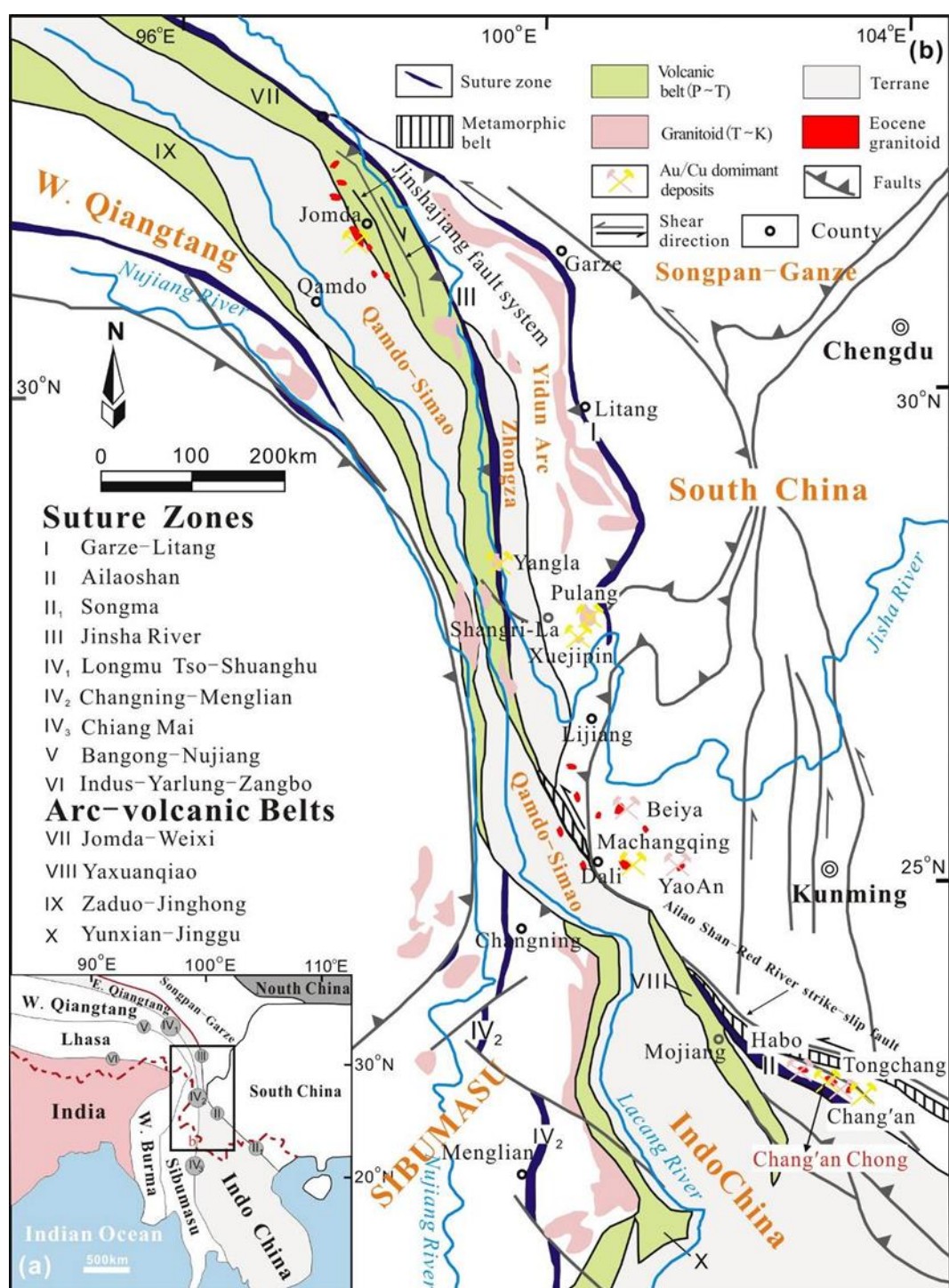

**Figure 1.** Distribution of principal continental blocks and sutures of southeast Asia (**a**). Tectonic framework of the Sanjiang domain in southwestern China, showing the major terranes, suture zones, arc volcanic belts and location of the Chang'an Chong Cu-Mo deposit and other primary porphyry-skarn Cu-Au deposits (**b**) (modified after [37,38]).

The area has experienced complex tectonic evolution and intense magmatic activity, providing reasonable physical and chemical conditions for the activation, migration and enrichment of various metal elements. Along the two sides of the Ailaoshan tectonic belt, a series of Au, Ag, Cu, Mo and other types of metal deposits are exposed in an NW-SE distribution. In the southern section of the structural belt, there is a place where the Chang'an gold deposit, Chang'anchong copper molybdenum deposit and Tongchang

copper deposit are concentrated. Scholars call it the Chang'an ore cluster area [39]; other typical deposits exposed in other parts of the tectonic belt include the Beiya Au deposit, Machangqing Cu-Mo deposit, Yao'an Au deposit and so on (Figure 2a).

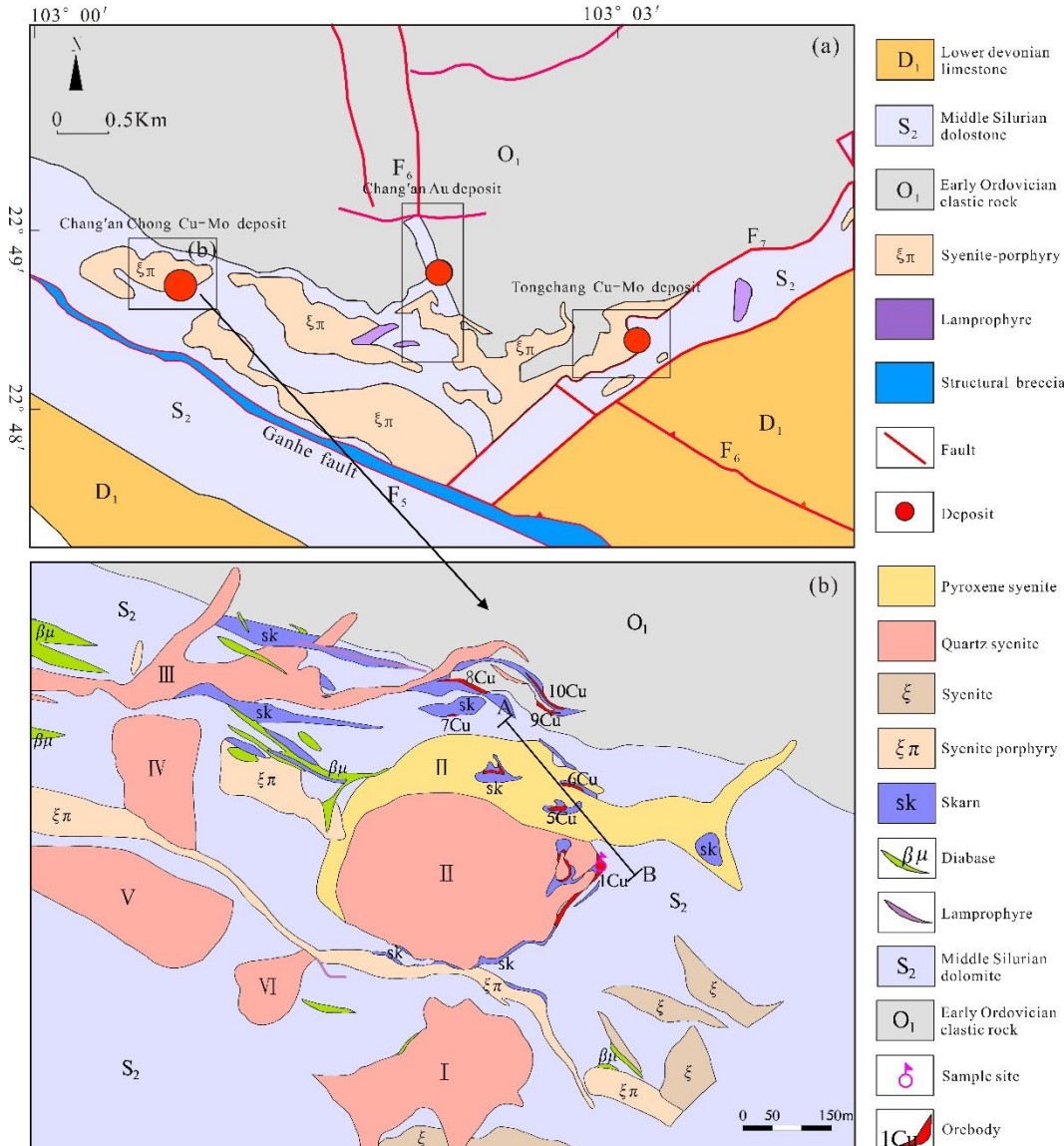

**Figure 2.** Simplified geological maps of the Chang'an Cu-Mo-Au ore cluster (**a**). Simplified geological maps of the Chang'an Chong Cu-Mo deposit (**b**) (modified after [9]).

## 3. Ore Deposit Geology

The Chang'an Chong Cu-Mo deposit is located in the middle of the Jinping fault block in the southern section of the Ailaoshan tectonic belt in southwestern China, with the Ailaoshan terrane in the northeast and Tengtiaohe fault in the southwest. The exposed strata in the mining area are the Ordovician ($O_1$) calcareous sandstone, quartz siltstone, quartz-feldspar conglomerate and middle Silurian ($S_2$) dolomite limestone. The structure of the mining area is complex. The main structure is a series of NW-trending compression-torsional faults, and the secondary structure is NE- and SN-trending wrench faults. The mining area has magmatic solid activity during the Himalayan period, which intruded along the Ganhe fault and is composed of 6 intrusive bodies, among which the No.II intrusive body has the largest scale (Figure 2b) and is most closely related to the Cu-Mo mineralization of skarn.

### 3.1. Geological Characteristics of Intrusive Bodies

The No.II intrusive body is located in the middle of the mining area. It is irregular and elliptical, approximately 1 km long and 0.7 km wide, and the outcropped area is about 0.18 km$^2$. The deep intrusive body dips and extends to the northeast, with a dip angle of ~37–49°. The intrusive body is columnar and intrudes into the Middle Silurian (S$_2$) dolomite and limestone, and the contact zone between the intrusive body and country rock which have been argillized is complex and uneven. According to the lithology difference, the intrusive body is divided into two parts: the northern part is mainly off-white fine-grained pyroxenite, and the southern part is primarily fleshy red porphyritic quartz syenite, and the relationship between them is an abrupt contact. Because xenoliths of the northern pyroxene syenite occur in the southern rock, it is inferred that the southern quartz syenite is younger. Lamprophyre dykes and diabase dykes are also found, but they are not closely related to mineralization.

### 3.2. Orebody Geological Characteristics

The Chang'an Chong Cu-Mo deposit has 21 industrial orebodies including 10 copper orebodies, 10 molybdenum orebodies and 1 iron orebody; all of them are in the contact metamorphic skarn and inside the alteration zone. The No. 1 copper orebody is adjacent to the No. 1 molybdenum in space, with similar orebodies in shape and the spatial distribution (Figure 3). The No. 1 copper orebody partially overlaps with the No. 1 molybdenum orebody, resulting in a sudden increase in molybdenum content, forming a copper-molybdenum mixed orebody, collectively known as the No. 1 main orebody. The rest of the orebodies are distributed near the main orebody, on a smaller scale.

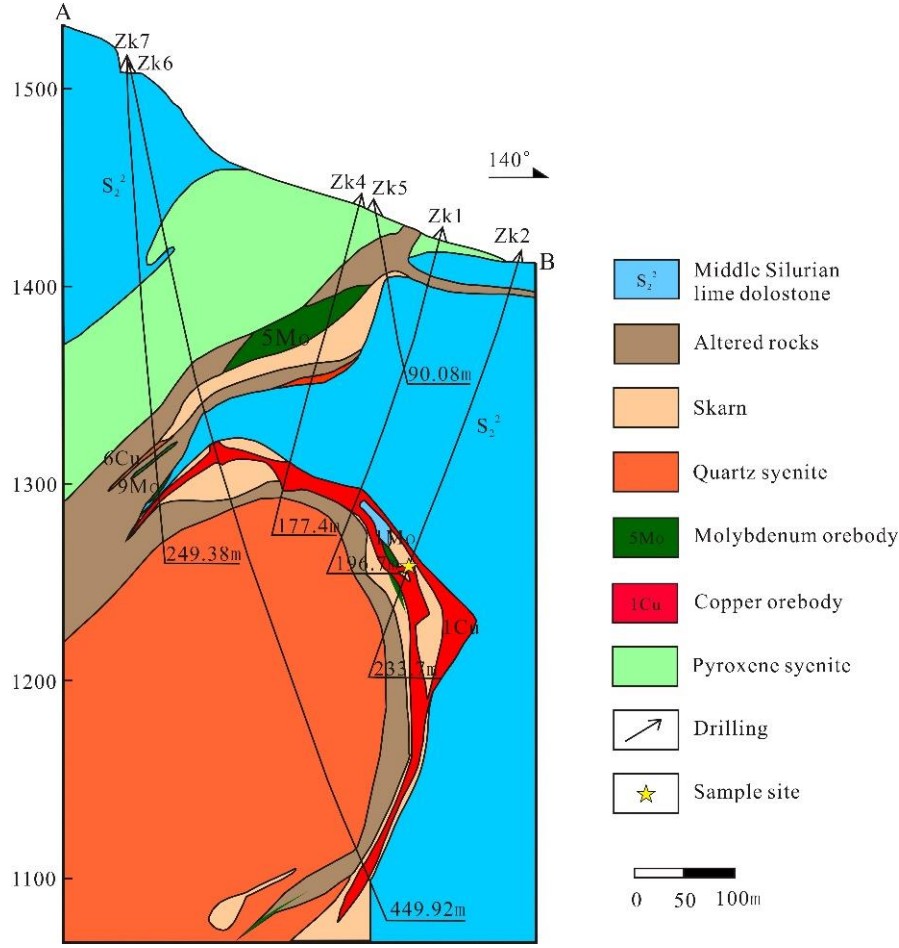

**Figure 3.** Geological sections along exploration line 4 of the Chang'an Chong Cu-Mo deposit.

The No. 1 copper orebody distribution is controlled by skarn and occurs in the skarn belt on the east side of the No. II intrusive body. The shape is a plate column, extending ~450 m along the strike, reaching ~370 m along with the dip, with an average thickness of ~6.5 m. The main metal minerals in the orebody are magnetite, chalcopyrite, pyrite, pyrrhotite, molybdenite, a small amount of marcasite and bornite. The copper content of the orebody is 0.3–13.5 wt.%, with an average of 1.45 wt.%. The gangue minerals are mainly quartz, feldspar, biotite, hornblende, pyroxene and garnet, and the minor minerals are forsterite, serpentine, phlogopite and chlorite. The No. 1 molybdenum orebody is located in the inside alteration zone of the outwards uplift on the south side of the No.II intrusive body. It is parallel to the No. 1 copper orebody, similar in shape, but small in size. It extends 50 m along the strike, reaches 490 m along the dip and has an average thickness of 4.76 m. The metallic minerals in the orebody are mainly molybdenite, chalcopyrite and pyrite, with a molybdenum content of ~0.001–0.5 wt.% and with an average of 0.24 wt.%. The gangue minerals are mainly albite, muscovite, chlorite and sericite. The ore textures consist of idiomorphic-hypidiomorphic granular, allotriomorphic granular, cataclastic and metasomatic, etc. The massive, disseminated, banded and inhomogeneous structures are the main ore structures.

### 3.3. Country Rock Alteration

Few endoskarns are distributed in the mining area, a small number of skarn veins can be seen in quartz syenite and a large number of exoskarns are developed in Middle Silurian dolomite and limestone. There are three types of the country rock alterations in the mining area: the inside alteration zone, skarn zone and outer alteration zone. Copper mineralization is closely related to the skarn zone, while molybdenum mineralization is closely associated with the inside alteration zone. (1) Inside alteration zone: thickness of 11.5~23 m; located at the edge of the intrusive body; the rocks are mainly silicified and sericite altered quartz monzonite and quartz syenite. The alteration is formed by contact metasomatism, assimilation and mixing, and hydrothermal alteration. (2) Skarn zone: thickness of 10.2~13.0 m; it is divided into the diopside skarn zone and forsterite skarn zone. The diopside skarn zone mainly comprises diopside, tremolite, actinolite and biotite. Diopside is replaced by tremolite and biotite. The forsterite skarn zone is mainly composed of forsterite, magnetite, carbonate and phlogopite, mostly located in diopside skarn. (3) Outer alteration zone: mainly serpentine marble and dolomitic marble.

### 3.4. Mineralization Stages

Under the field expedition of orebodies, mineral symbiotic association and microscopic identification, the mineralization can be divided into 3 stages and 6 sub-stages (Figures 4 and 5). (1) The skarn stage can be divided into 3 sub-stages: (a) early skarn sub-stage: mainly composed of anhydrous skarn minerals such as garnet, forsterite and diopside; (b) late skarn sub-stage: water-bearing silicate minerals such as actinolite, tremolite and epidote are mainly formed, accompanied by a large number of magnetite; (c) oxide sub-stage: mainly mica minerals, quartz and a small number of pyrrhotite and molybdenite are formed. (2) The sulfide stage can be divided into 3 sub-stages: (a) Fe-Mo sulfide sub-stage: dominated by a large number of molybdenite, pyrite and quartz, accompanied by a small number of pyrrhotite, chlorite and serpentine; (b) Fe-Cu sulfide sub-stage: chalcopyrite, bornite and quartz are dominant, with a small number of chlorite and serpentine; (c) Pb-Zn sulfide sub-stage: mainly galena and sphalerite. (3) Supergene stage: mainly limonite.

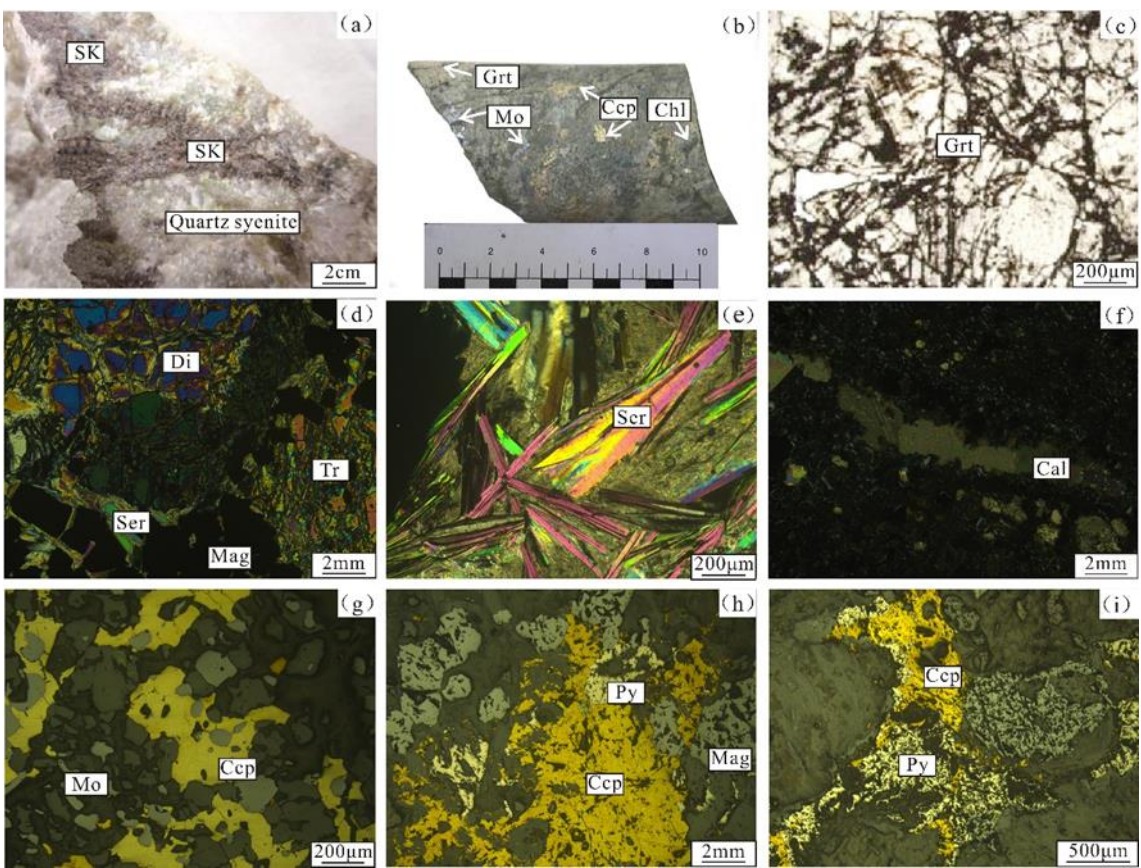

**Figure 4.** Photographs of typical ores from the Chang'an Chong Cu-Mo deposit. (**a**) Skarn intercalated quartz syenite; (**b**) garnet coexisting with chalcopyrite and molybdenite; (**c**–**f**) minerals from the skarn stage; (**g**–**i**) minerals from the sulfide stage. Cal-Calcite; Ccp-Chalcopyrite; Chl-Chlorite; Di-Diopside; Mag-Magnetite; Mo-Molybdenite; Py-Pyrite; Ser-Sericite; Tr-Tremolite.

| Minerals | Skarn stage | | | Sulfide stage | | | Supergene stage |
|---|---|---|---|---|---|---|---|
| | Early skarn sub-stage | Late skarn sub-stage | Oxide sub-stage | Fe–Mo Sulfide sub-stage | Fe–Cu Sulfide sub-stage | Pb–Zn Sulfide sub-stage | |
| Garnet | ⬛ | | | | | | |
| Diopside | ⬛ | | | | | | |
| Forsterite | ⬛ | | | | | | |
| Actinolite | | ⬛ | | | | | |
| Tremolite | | ⬛ | | | | | |
| Epidote | | ⬛ | | | | | |
| Phlogopite | | | ⬛ | | | | |
| Sericite | | | ⬛ | | | | |
| Chlorite | | | | ⬛ | | | |
| Serpentite | | | | ⬛ | | | |
| Magnetite | | ⬛ | | | | | |
| Quartz | | | ⬛ | | | | |
| Molybdenite | | | | ⬛ | | | |
| Pyrrhotite | | | | ⬛ | | | |
| Arsenopyrite | | | | ⬛ | | | |
| Pyrite | | | | ⬛ | | | |
| Chalcopyrite | | | | | ⬛ | | |
| Bornite | | | | | ⬛ | | |
| Sphalerite | | | | | | ⬛ | |
| Calcite | | | | | | ⬛ | |
| Galena | | | | | | ⬛ | |
| Limonite | | | | | | | ⬛ |

**Figure 5.** Formation sequence of main minerals of the Chang'an Chong Cu-Mo deposit.

## 4. Samples and Analytical Methods

### 4.1. Sample Collection

The analyzed garnets and samples of the Chang'an Chong Cu-Mo deposit were taken from the ZK2 borehole (Figure 3). All samples were relatively fresh, and free from weathering and alteration. The studied samples were made into thin sections for detailed mineralogical observation by polarized-light microscopy to determine the minerals' composition, structure and symbiotic relationship. Then, samples were cut from CA-05, CA-08, CA-09, CA-15 and CA-18 to make a 0.04 mm probe sheet for analysis: U-Pb dating of garnet was carried out for CA-05, and in-situ S isotope analysis of sulfide was carried out for CA-05, CA-08, CA-09, CA-15 and CA-18.

### 4.2. Analytical Methods

The U-Pb dating of garnet was completed in the State Key Laboratory of Ore Deposit Geochemistry, Institute of Geochemistry, Chinese Academic of Science.

Before testing, appropriate samples were selected to grind probe pieces to a thickness of ~0.04 mm. Andradite garnet particles with a good crystal structure were picked by the results of the major element text and microscope. When testing, mineral cracks, inclusions and other impurities were avoided to reduce the impact of common Pb.

The Thermo Element XR high-resolution magnetic mass spectrometry (HR-SF-ICP-MS) and excimer laser ablation system (GeoLasPro 193 nm) were used for the analysis. The laser ablation spot diameter was set to 32 μm and the laser energy density was set to 3 J/cm$^2$, denudation frequency was 5 Hz, He was used as a carrier gas for ablation material (0.45 L/s) and Ar was used as the auxiliary gas. The sampling time of single point denudation was ~90 s, including a background acquisition time of the 20 s, laser denudation time of 30 s and cleaning time of 40 s. The samples were analyzed twice according to the sequence of standards NIST SRM 612, 91500, Willsboro and QC04 before the test. After each analysis of 10-15 sample sites, the above standard samples were analyzed twice. At the end of the test, the samples were analyzed twice in the reverse sequence as at the beginning of the test. In the analysis, standard zircon 91500 (1062 Ma) was used as the main standard sample, and garnet QC04 was used as the quality control, which obtained a lower intercept $^{206}Pb/^{238}U$-weighted mean age 132.8 ± 1.9 Ma (MSWD = 0.6, *n* = 14), which was consistent with the recommended value within the error range (130 ± 1 Ma; [14]). The analysis process and detailed steps are referred to in [13,40], the detailed description of QC04 and Willsboro is referred to in [13] and the U-Th-Pb isotope ratio of zircon standard sample 91500 is referred to [41]. The analysis results were processed using the ICPMSDataCal program [42,43], and U-Pb age calculations and plots were calculated using Isoplot 4.5 program [44], with an error of 1σ between isotope ratios and ages.

An in-situ S isotope analysis of sulfide was performed in the MNR Key Laboratory of Metallogeny and Mineral Assessment, Institute of Mineral Resources, Chinese Academy of Geological Sciences. The ArF excimer laser generator generated a 193 nm deep-ultraviolet beam, which was focused on the surface of sulfide through the homogenized optical path. The laser beam spot diameter was generally selected as 33 μm, the denudation frequency was 10 Hz and the denudation time was 40 s. High purity helium gas as a carrier gas was mixed with argon gas and nitrogen gas and then entered the mass spectrometer. Direct testing was performed to obtain standard samples and the sample $^{34}S/^{32}S$ ratio, using the external standard correction method (SSB method) to calculate the $\delta^{34}S_{CDT}$ value. The standard samples were international sulfide standard NBS-123 sphalerite and laboratory internal standard WS-1 pyrite [45]. The international recommended value of $\delta^{34}S$ for NBS-123 sphalerite was +17.1‰, WS-1 pyrite value of $\delta^{34}S$ measured by the gas mass spectrometer was +0.9‰ and that measured by the ion microprobe was +1.1‰ ± 0.2‰. When using WS-1 as the standard calibration, the value of $\delta^{34}S$ for NBS-123 +17.0‰ ± 0.5‰, indicating that the matrix effect among different sulfides was not obvious. In this test, WS-1 was used as the standard sample for correction, and the accuracy of the 2σ analysis was ±0.3‰

## 5. Results

### 5.1. U-Pb Dating of Garnet

The main isotope signals obtained by LA-SF-ICP-MS include $^{206}$Pb, $^{207}$Pb, $^{208}$Pb, $^{232}$Th and $^{238}$U. The signal acquisition time is ~25 s, and the data are overall stable (Figure 6a). In addition, except for a small number of inclusions in pyroxene, no large fluid inclusions are found in garnet during this study. These features suggest that the U in garnet is mainly derived from the mineral lattice; thus, garnet U-Pb age represents the formation age of the mineral.

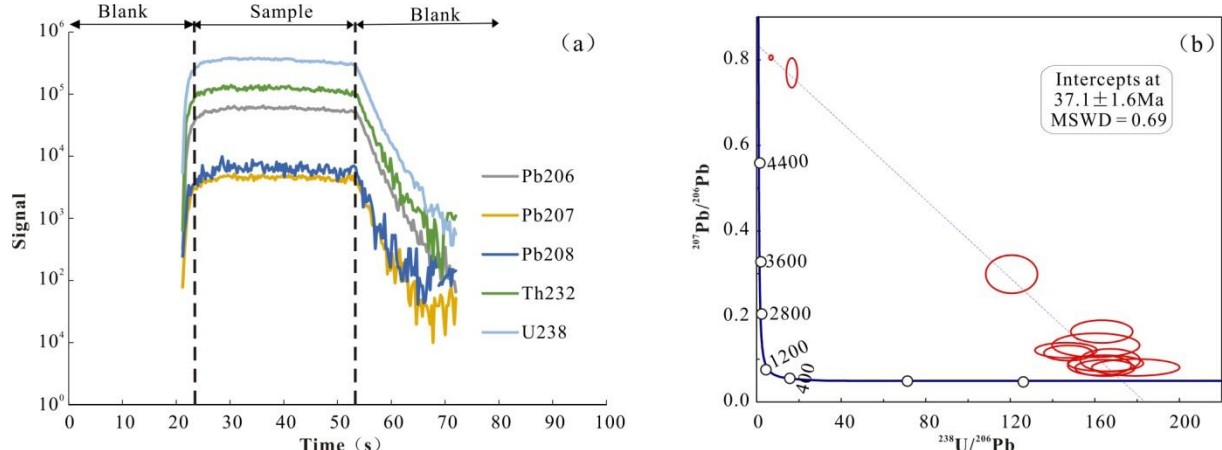

**Figure 6.** Date curves (**a**) and result (**b**) of LA-SF-ICP-MS U-Pb dating of garnet from the Chang'an Chong Cu-Mo deposit.

In this study, 18 U-Pb isotope measuring points are analyzed for CA-05. Except for the poor data signal and instability caused by small inclusions or other cracks that may be eroded in the testing process, a total of 15 effective data points are obtained (Figure 6b, Table 1). With the Pb content of $0.06 \times 10^{-6}$–$2.63 \times 10^{-6}$, the average value is $0.42 \times 10^{-6}$; the Th content is $0.75 \times 10^{-6}$–$1.84 \times 10^{-6}$, and the average value is $1.26 \times 10^{-6}$; the U content is $2.87 \times 10^{-6}$–$5.92 \times 10^{-6}$, the average value is $4.46 \times 10^{-6}$; the $^{207}$Pb/$^{206}$Pb ratio varies from 0.07 to 0.80, the $^{207}$Pb/$^{235}$U ratio varies from 0.06 to 16.30 and the $^{206}$Pb/$^{238}$U ratio ranges from 0.01 to 0.15. Due to the high content of common Pb, common Pb is corrected by using the Tera–Wasserburg Concordia diagram anchored at 4950 Ma [13,40], and the lower intercept $^{206}$Pb/$^{238}$U age of garnet is 37.1 ± 1.6 Ma (MSWD = 0.69, *n* = 15).

**Table 1.** LA-ICP-MS U-Pb dating of the garnet in the Chang'an Chong Cu-Mo deposit.

| Sample No. | ppm | | | Th/U | Isotope Ratio | | | | | |
| --- | --- | --- | --- | --- | --- | --- | --- | --- | --- | --- |
| | Pb | Th | U | | $^{207}$Pb/$^{206}$Pb | 1 Sigma | $^{207}$Pb/$^{235}$U | 1 Sigma | $^{206}$Pb/$^{238}$U | 1 Sigma |
| CA05-01 | 0.06 | 1.09 | 4.89 | 0.22 | 0.1202 | 0.0263 | 0.0620 | 0.0111 | 0.0056 | 0.0004 |
| CA05-02 | 0.08 | 0.96 | 5.10 | 0.19 | 0.1502 | 0.0239 | 0.1070 | 0.0123 | 0.0068 | 0.0004 |
| CA05-03 | 0.07 | 1.19 | 5.36 | 0.22 | 0.0705 | 0.0126 | 0.0664 | 0.0112 | 0.0061 | 0.0003 |
| CA05-05 | 0.13 | 1.29 | 5.92 | 0.22 | 0.3394 | 0.0508 | 0.3414 | 0.0407 | 0.0083 | 0.0006 |
| CA05-06 | 0.07 | 1.40 | 4.88 | 0.29 | 0.1381 | 0.0323 | 0.0754 | 0.0117 | 0.0061 | 0.0004 |
| CA05-07 | 0.08 | 1.50 | 5.17 | 0.29 | 0.1826 | 0.0289 | 0.1389 | 0.0169 | 0.0061 | 0.0004 |
| CA05-08 | 0.09 | 1.34 | 5.18 | 0.26 | 0.1526 | 0.0237 | 0.1137 | 0.0134 | 0.0068 | 0.0005 |
| CA05-09 | 1.18 | 1.63 | 5.20 | 0.31 | 0.7521 | 0.0347 | 6.3694 | 0.6847 | 0.0600 | 0.0062 |
| CA05-10 | 2.63 | 0.82 | 4.70 | 0.18 | 0.8023 | 0.0270 | 16.3023 | 1.4166 | 0.1468 | 0.0127 |
| CA05-11 | 0.24 | 1.69 | 4.30 | 0.39 | 0.1180 | 0.0233 | 0.0803 | 0.0152 | 0.0060 | 0.0003 |
| CA05-12 | 0.20 | 1.84 | 4.45 | 0.41 | 0.1208 | 0.0317 | 0.0727 | 0.0139 | 0.0062 | 0.0004 |
| CA05-14 | 0.33 | 1.03 | 2.87 | 0.36 | 0.1367 | 0.0281 | 0.0968 | 0.0146 | 0.0079 | 0.0006 |
| CA05-15 | 0.37 | 1.39 | 2.90 | 0.48 | 0.1365 | 0.0253 | 0.1286 | 0.0178 | 0.0074 | 0.0005 |
| CA05-16 | 0.33 | 1.02 | 3.00 | 0.34 | 0.1653 | 0.0412 | 0.1138 | 0.0186 | 0.0062 | 0.0005 |
| CA05-18 | 0.43 | 0.75 | 2.97 | 0.25 | 0.1435 | 0.0243 | 0.1285 | 0.0167 | 0.0074 | 0.0004 |

### 5.2. In-Situ S Isotopes

In this study, in-situ S isotope tests are carried out on different types of pyrite and chalcopyrite in the Chang'an Chong Cu-Mo deposit, including disseminated pyrite (PyI), metallic sulfide pyrite (PyII) and chalcopyrite (Ccp) in the main mineralization period. Disseminated pyrite (PyI) is mostly distributed in the diopside skarn, showing fine spot disseminated (0.01–0.35 mm), spotted (5–10 mm), fibrous and other structural forms; metallic sulfide pyrite (PyII) is mostly distributed in the internal alteration zone, often distributed around the chalcopyrite and molybdenite, generally less than 1 mm in width. Among them, PyI, PyII and Ccp are tested at 16 points, 2 points and 4 points, respectively (Figure 7, Table 2). The values of $\delta^{34}S$ for the two types of pyrites are relatively concentrated from 1.9‰ to 3.8‰ (average 2.49‰, range 1.9‰), including the values of $\delta^{34}S$ for PyI of 1.9‰–2.9‰ (average 2.35‰, 1.0‰), PyII ranging from 3.4‰ to 3.8‰ (average 3.60‰, range 0.4‰) and the values of $\delta^{34}S$ for PyI are lower than PyII. The values of $\delta^{34}S$ for Ccp in the main mineralization period range from 0.0‰ to 0.9‰ (average 0.55‰, range 0.9‰) (Figure 8).

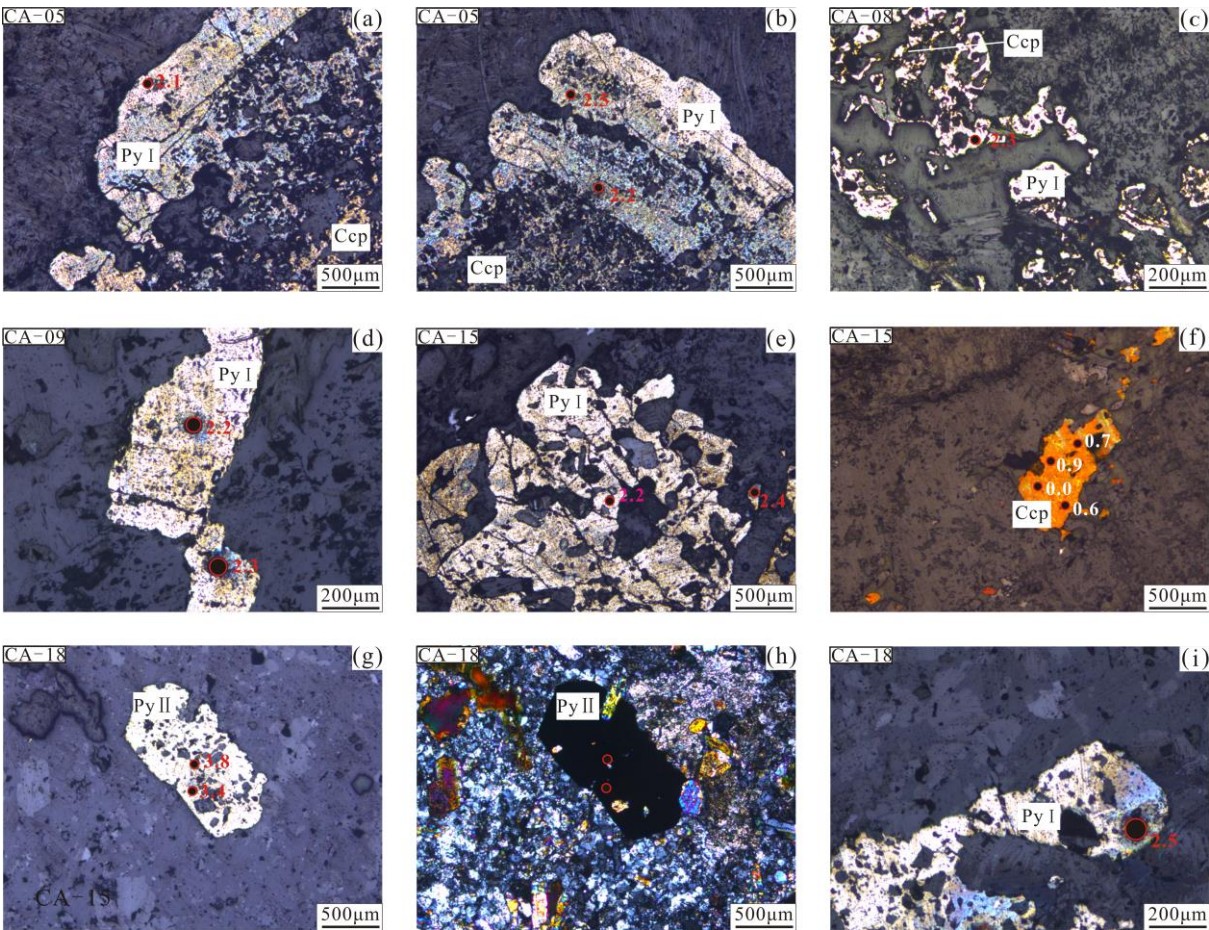

**Figure 7.** Micrographs of the sulfide minerals in the Chang'an Chong Cu-Mo deposit. (**a**,**b**) CA-05: the values of $\delta^{34}S$ for PyI; (**c**) CA-08: the values of $\delta^{34}S$ for PyI; (**d**) CA-09: the values of $\delta^{34}S$ for PyI; (**e**,**f**) CA-15: the values of $\delta^{34}S$ for PyI, Ccp ; (**g**–**i**) CA-18: the values of $\delta^{34}S$ for PyI, PyII.

**Table 2.** $^{34}$S isotope analysis results of the Chang'an Chong Cu-Mo deposit.

| Sample No. | Mineral | Stage | $^{34}$S$_{V-CDT}$‰ | Sample No. | Mineral | Stage | $^{34}$S$_{V-CDT}$‰ |
|---|---|---|---|---|---|---|---|
| CA-05-1 | pyrite | PyI | 2.1 | CA-15-1 | pyrite | PyI | 1.9 |
| CA-05-2 | pyrite | PyI | 2.5 | CA-15-2 | pyrite | PyI | 2.2 |
| CA-05-3 | pyrite | PyI | 2.2 | CA-15-3 | pyrite | PyI | 2.4 |
| CA-08-1 | pyrite | PyI | 2.2 | CA-15-4 | pyrite | PyI | 2.5 |
| CA-08-2 | pyrite | PyI | 2.3 | CA-18-1 | pyrite | PyII | 3.8 |
| CA-08-3 | pyrite | PyI | 2.4 | CA-18-2 | pyrite | PyII | 3.4 |
| CA-08-4 | pyrite | PyI | 2.5 | CA-18-3 | pyrite | PyI | 2.5 |
| CA-09-1 | pyrite | PyI | 2.6 | CA-15-5 | chalcopyrite | main mineralization period | 0.6 |
| CA-09-2 | pyrite | PyI | 2.2 | CA-15-6 | chalcopyrite | main mineralization period | 0.0 |
| CA-09-3 | pyrite | PyI | 2.9 | CA-15-7 | chalcopyrite | main mineralization period | 0.9 |
| CA-09-4 | pyrite | PyI | 2.3 | CA-15-8 | chalcopyrite | main mineralization period | 0.7 |

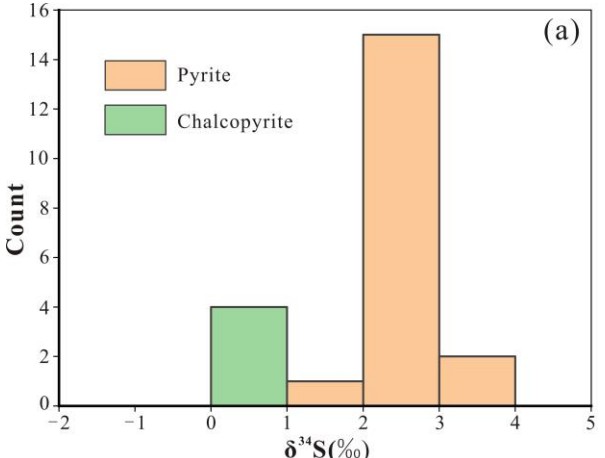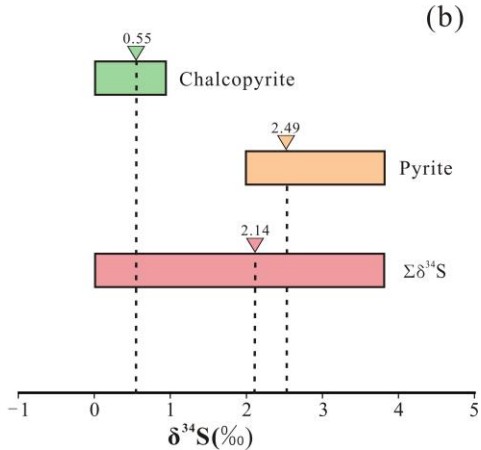

**Figure 8.** In-situ S isotope distribution diagram of the Chang'an Chong Cu-Mo deposit. (**a**) frequency histogram of $\delta^{34}$S; (**b**) average of $\delta^{34}$S.

## 6. Discussion

### 6.1. Constraints on the Timing of Skarn Formation and Mineralization

The study of the metallogenic epoch is helpful to understand the temporal and spatial distribution characteristics and evolution of ore deposits. Generally, the study of the metallogenic age of skarn deposits can be limited by zircon U-Pb dating of granite, porphyry and other intrusive bodies. The most accurate method is directly selecting metallic minerals for isotopic dating, such as molybdenite Re-Os and scheelite Sm-Nd. In recent years, through the study of skarn mineralization stages, elaborately, the age of garnet has been used to constrain its mineralization [16–24].

U-Pb dating of garnet containing skarn in the Chang'an Chong Cu-Mo deposit shows that the age of garnet is 37.1 ± 1.6 Ma (MSWD = 0.69, *n* = 16), which represents the age of skarn in this period, and is consistent with the LA-ICP-MS U-Pb age of zircon from the No. II quartz syenite (34.5 ± 0.3 Ma, [9]). According to the metallogenic characteristics of the Chang'an Chong Cu-Mo deposit, garnet runs through the skarn stage, and the Cu-Mo orebodies formed in the late skarn stage, and are embedded in garnet, pyroxene, tremolite, actinolite and other minerals. It can be considered that the age of the skarn and the Cu-Mo orebodies are nearly identical, which also indirectly limited the metallogenic age of the Cu-Mo deposit. Therefore, it is believed that the magmatic emplacement, skarn and Cu-Mo metallogenic age of ~37 Ma are consistent, representing the products of the same mineralization event, which indicate that the U-Pb dating of garnet is reliable.

Previous studies have dated the metallogenic age of the Tongchang Cu-Mo deposit and Chang'an Au deposit, respectively, which belong to the Chang'an ore cluster. The zircon U-Pb ages of fine-grained syenite from the Tongchang Cu-Mo deposit are 36.0 ± 0.19 Ma,

syenite porphyry is 34.60 ± 0.20 Ma [46], monzonite porphyry is 35.1 ± 0.3 Ma [47] and molybdenite Re-Os isochron is 34.4 ± 0.5 Ma [43]. The LA-ICP-MS U-Pb ages of the fine-grained syenite and syenite porphyry in the Chang'an Au deposit are 32.5 ± 0.1 Ma and 33.0 ± 0.1 Ma, respectively; the $^{39}$Ar-$^{40}$Ar plateau age of biotite in lamprophyre is 35.62 ± 0.16 Ma; and the isochron age is 35.27 ± 0.74 Ma [48]. The zircon U-Pb age of granite porphyry vein is 37.1 ± 0.5 Ma [8], and the molybdenite isochron age is 34.5 ± 0.7 Ma [49]. The U-Pb dating of garnet in the Chang'an Chong Cu-Mo deposit is 37.1 ± 1.6 Ma. Compared with the U-Pb age of zircon from the ore-bearing porphyry and the Re-Os age of molybdenite, the U-Pb age of garnet is consistent with them within the error range, indicating that they were the products of the same mineralization event. This is also consistent with the rock-forming age and metallogenic epochs of the Daping Au deposit, Machangqing Cu deposit and Beiya Au deposit related to alkaline porphyry in the region, indicating that the mineralization peak period of the porphyry-skarn Cu-Mo-Au deposit distributed in the Ailaoshan-Red River fault zone is ~32–38 Ma. Therefore, the accurate U-Pb dating of garnet can effectively confirm the age of the porphyry-skarn deposit. It is a new geochronology method that supports the metallogenic mechanism.

### 6.2. Ore-Forming Material Sources

The $\delta^{34}$S values of sulfide minerals in the Chang'an Chong Cu-Mo deposit measured by the single mineral particle powder method range from 0.2‰ to 1.5‰ [9]: the values are all positive and vary in a small field. In this paper, the $\delta^{34}$S values of metal sulfide are obtained by an in-situ micro-area measurement with values ranging from 0.0‰ to 3.8‰ (average 2.14‰), which is broader than those measured by the single mineral particle powder method. The in-situ test method can more effectively measure the $\delta^{34}$S value of a single sulfide mineral, and can be obtained more accurately, to more effectively trace the source of sulfur in metallogenic materials [50,51]. The sulfide assemblage of the Chang'an Chong Cu-Mo deposit is relatively simple, mainly including chalcopyrite, pyrrhotite, pyrite and sphalerite. Combined with the fact that sulfate minerals are not found in the mining area, the $\delta^{34}$S value of sulfide can represent the total sulfur isotopic composition of the ore-forming hydrothermal fluids [50,52].

Generally, the $\delta^{34}$S values of pyrites in the two periods change slightly, but the overall difference is not significant. It is similar to the sulfur isotope composition of the magma, and slightly enriched $\delta^{34}$S. The $\delta^{34}$S value of chalcopyrite in the metallogenic period is mainly concentrated near 0‰, so it can be considered that the sulfur of ore-forming fluid comes from the magmatic-hydrothermal fluid. The specific results are shown in Table 2. The average $\delta^{34}$S value of skarn disseminated pyrite (PyI) in this deposit is 2.35‰, and the sulfur isotope composition is stable and uniform. The variation range of $\delta^{34}$S values is narrow, showing similar source characteristics of the S isotopes, which are within the range of S isotope compositions of deep source magma (0 ± 3‰) [50]. The S isotopic composition of porphyry closely related to gold polymetallic mineralization in western Yunnan was studied by the predecessors. The $\delta^{34}$S value of alkaline porphyry is 1.0‰~2.5‰ [53], which is consistent with the S isotopic composition of pyrite (PyI) in the region, indicating that the sulfur in the area has the characteristics of deep source magma sulfur and is directly related to the nearby porphyry.

The $\delta^{34}$S values of pyrite (PyII) from metal sulfide veins are 3.8‰ and 3.4‰, with an average value of 3.60‰. Generally, mantle-derived sulfur is the primary source. The $\delta^{34}$S values of chalcopyrite in the main mineralization period are concentrated near the zero value, and slightly enriched, indicating that the ore-forming materials in the main mineralization period were mainly mantle-derived (Figure 9).

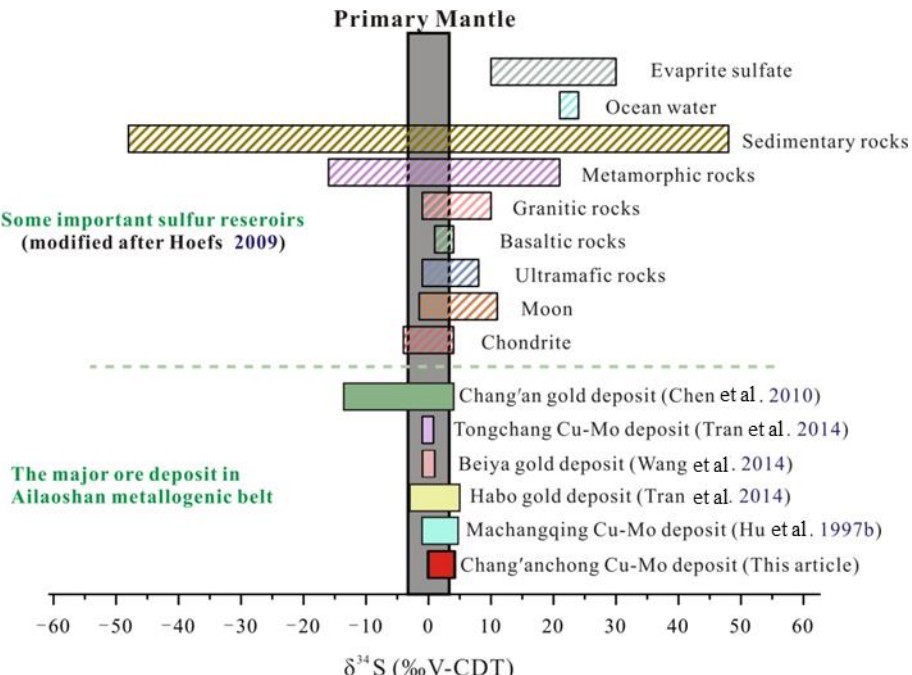

**Figure 9.** S isotope composition diagram of typical deposits in the Ailaoshan metallogenic belt [54–58].

*6.3. Metallogenic Mechanism and Geodynamic Background*

The Sanjiang region has experienced three stages of main collision convergence (~65–41 Ma), late collision transition (~40–26 Ma) and post-collision extension (~25–0 Ma) since the Cenozoic India-Eurasia collision, corresponding to when the main collision orogenic environment, late collision strike-slip environment and post-collision extension environment were formed, respectively. During the late collision transition period of ~40–26 Ma, a series of regional-scale thrust-nappe-strike-slip fault systems were formed along the boundaries of different terrains in the Sanjiang region, which adjusted and absorbed the stress deformation caused by the India-Eurasia collision (Figure 1). At the same time, many potassium-rich alkaline igneous rocks were exposed along with these Cenozoic strike-slip fault systems, which were considered to be controlled by the tectonic stress field of the Cenozoic intracontinental strike-slip transition [59–62]. The Chang'an Chong Cu-Mo skarn deposit is located in the southern section of the Jinshajiang-Red River alkaline intrusive bodies. The quartz syenite intrusion is located in the Jinping micro-block sandwiched by the Ailaoshan-Red River fault and the Tengtiaohe fault, which is adjacent to the Red River strike-slip fault in space. From the obtained U-Pb age of garnet (37.1 ± 1.6 Ma) and previous zircon U-Pb age (34.5 ± 0.3 Ma, [9]) of the Chang'an Chong Cu-Mo deposit, the metallogenic age is approximately ~37 Ma, which is generally in the late collision transition period (~40–26 Ma) in Sanjiang region [61]. Therefore, it can be determined that the Chang'an Chong Cu-Mo deposit was formed in a late collision strike-slip environment from two aspects of geotectonic location and metallogenic age, which is consistent with the timing of skarn-forming and tectonic background of Chang'an granite porphyry (~37 Ma, [8]), Habo monzonite porphyry (~36 Ma, [63]) and Tongchang monzonitic granite porphyry (~35 Ma, [47]). Both of them were formed in the intracontinental strike-slip environment [64,65].

## 7. Conclusions

(1) The U-Pb dating of garnet in the Chang'an Chong Cu-Mo deposit is 37.1 ± 1.6 Ma, showing that the age of skarn was formed in the Late Eocene. There is no doubt that the mineralization stage is nearly consistent with skarn. It can represent an ore-forming state.

(2) The $\delta^{34}$S value of sulfide in the Chang'an Chong Cu-Mo deposit ranges from 0.0‰ to 3.8‰, indicating that the ore-forming materials are mainly mantle-derived.

(3) The Chang'an Chong Cu-Mo deposit was formed in the intracontinental strike-slip environment.

**Author Contributions:** Conceptualization, B.S., Y.L. and Y.Y.; methodology, Y.L. and L.Y.; formal analysis, B.S., Y.L. and G.C.; investigation, B.S., Y.L. and G.C.; resources, Y.L. and Y.Y.; data curation, Y.L.; writing—original draft preparation, B.S., Y.L. and G.C.; writing—review and editing, Y.L. and L.Y.; supervision, Y.L. and Y.Y.; project administration, Y.L. and Y.Y.; funding acquisition, Y.L. and Y.Y. All authors have read and agreed to the published version of the manuscript.

**Funding:** This research was supported by Yunnan Major Scientific and Technological Projects (Grant No. 202202AG050006); The Key R&D Program of Yunnan Province (Grant No. 202103AQ100003).

**Data Availability Statement:** All data generated or analyzed during this study are included in this published article.

**Acknowledgments:** The sample testing was greatly assisted by the Institute of Geochemistry Chinese Academy of Sciences and the Key Laboratory of Metallogeny and Mineral Assessment, Institute of Mineral Resources, Chinese Academy of Geological Sciences. Thanks to the anonymous reviewers for their valuable comments on this paper.

**Conflicts of Interest:** The authors declare no conflict of interest.

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
