# Peer review of "Metallogenic Mechanism and Geodynamic Background of the Chang’an Chong Cu-Mo Deposit in Southern Ailaoshan Tectonic Belt: New Evidence from Garnet U-Pb Dating and In-Situ S Isotope"

_minerals, doi:10.3390/min12111389_

Round 1

Reviewer 1 Report

Comments to Sun et al. Ms: minerals-1946620

Sun et al. Presents a garnet U-Pb age and in situ S isotopes of sulfide minerials in Changan Chong Cu-Mo deposit. The data quality is high and the geolocical meaning is significant. From this poit of view, this ms is suitable to Minerials.

Howerver, the english of this ms is not very good, and the discussion part is too wordy and repeatied for many times. So major revision of this ms is needed, and can be accepted after cut many useless sentences.

Some detail comments are listed as follows:

1 line52-52 the studied Co-Mo deposit is really small to be worthly studied.

2 line 55-56. why this place is a good place for finding big ore deposit? Any direct evidences? Geophysical evidences?

3 line 73-76L delet it

4 line 103: adakite; delete series

5 line 105: delete formed in the same peirod

6 line 107-108: what is the differences between these K-rich calc-alkalic rocks to 41-32ma high k-Si rocks?

7 line 118-120: please reorganized this sentence. It is really bad presentation here.

8 line 126-127: change The mining area is mainly exposed in “ to “The exposed strata in mining area are

9 line 143: there is no pyroxineite in the geological map, why?

10 line 144-147: is there any field pictures about this presentation?

11 line 153-154:After years of exploration, the Chang'an Chong Cu-Mo deposit has found 21 industrial orebodies, 10 copper orebodies, 10 molybdenum orebodies, and 1 iron orebody” to “The Chang'an Chong Cu-Mo deposit has 21 industrial orebodies including 10 copper orebodies, 10 molybdenum orebodies, and 1 iron orebody.”

12 160-161: what is the meaning of “The rest of the orebodies are produced in the main orebody”

13 182-184: wrong discription about the skarn deposit. For skarn deposit, endoskarn and exoskarn need classified at first. How is the distences from the diospide riched zone and the frosterite riched zone to the intrusions?

14 line 189-192: please show representive pictures of the discribed phenomials.

15 line 197: previous discription do not have garnet.

16 line 198: no magnitite here?

17 line 200: where is the forsterite?

18 line 203: what is the prophyrite?

19 line 204-205: quartze-calcite stage is not discripe here, but show in fig4C?

20 figure 5 need to be redraw.

22 line 233: HR-ICPMS or SF-ICPMS?

23 line 248:how about the willsboro garnet age?

24 line 270: elements sequence is pb 206,207,208, th232 and u238;

25 line 271: according to fig6. the laser ablation time is 40s? not 50s?

26 line 276: only 18 spots for garnet u-pb dating looks too litter to get a good isochrone age.

27 line 286:fig.6a; this is not the reprsentive garnet signatures, because the U signal looks very high. Please show a typical garnet LA signal

28 line 292-293:the two types of pyrites do not discribed in the sample discription part.

29 line294: why the in sito S isotopes analysis of different sulfide is so differencent, for pyrite1, 16 points, but for py2 and Ccp are only 2 and 4 spots? 2 and 4 is not enough for represent the S isotopes signals for their sources

30 line 315: change to the age of garnet is

31 line 327: “ore-concerntrated area” to “ore cluster”

32 line 340-341:looks like the fault  age is younger than ore forming age? 32-28?

33 line 351-356: useless sentences. Shot it in simple way.

34:line 359-360: wrong statement. the sulfide minerals is not the prof of redox state of the fluids. it changed very much. so at the begining of the magma source redox state is importent.

35 line 380-381: fractionation is another explanation, how can you exclude this posiblility?

36 line 392:any evidences for atmospheric water?

Reviewer 2 Report

It is interesting to use garnet U-Pb dating to constrain the ore-forming age and aim ore-forming magmatic rocks combined with S and lead isotopes.

This Manuscript “Metallogenic mechanism and geodynamic background of the Chang′an Chong Cu-Mo deposit in southern Ailaoshan Tectonic belt: New evidence from garnet U-Pb dating and in-situ S isotope” offer an good U-Pb age and some S isotopes to constrain the genesis of Chang’an Chong Cu-Mo deposit. It is useful and welcome. However, there are some misunderstanding in S isotopes, and such data need to be explained more reasonable.

Some detail comments in PDF will help to modified this Manuscript.
